# Feasibility Analysis of a Membrane Desorber Powered by Thermal Solar Energy for Absorption Cooling Systems

**Jonathan Ibarra-Bahena** [1], **Eduardo Venegas-Reyes** [1], **Yuridiana R. Galindo-Luna** [2], **Wilfrido Rivera** [2], **Rosenberg J. Romero** [3], **Antonio Rodríguez-Martínez** [3] and **Ulises Dehesa-Carrasco** [4],*

[1] Instituto Mexicano de Tecnología del Agua, Paseo Cuauhnáhuac 8532, Colonia Progreso, Jiutepec 62550, Morelos, Mexico; jibarra@ier.unam.mx (J.I.-B.); eduardo_venegas@tlaloc.imta.mx (E.V.-R.)

[2] Instituto de Energías Renovables, Universidad Nacional Autónoma de México, Privada Xochicalco S/N, Col. Centro 62580, Temixco, Morelos, Mexico; ygalu@ier.unam.mx (Y.R.G.-L.); wrgf@ier.unam.mx (W.R.)

[3] Centro de Investigación en Ingeniería y Ciencias Aplicadas, Universidad Autónoma del Estado de Morelos, Av. Universidad 1001, Col. Chamilpa, Cuernavaca 62209, Morelos, Mexico; rosenberg@uaem.mx (R.J.R.); antonio_rodriguez@uaem.mx (A.R.-M.)

[4] Consejo Nacional de Ciencia y Tecnología-Instituto Mexicano de Tecnología del Agua, Paseo Cuauhnáhuac 8532, Colonia Progreso, Jiutepec 62550, Morelos, Mexico

* Correspondence: udehesaca@conacyt.mx

**Abstract:** In absorption cooling systems, the desorber is a component that separates the refrigerant fluid from the liquid working mixture, most commonly completed by boiling separation; however, the operation temperature of boiling desorbers is generally higher than the low-enthalpy energy, such as solar, geothermal, or waste heat. In this study, we used a hydrophobic membrane desorber to separate water vapor from an aqueous LiBr solution. Influencing factors, such as the $H_2O$/LiBr solution and cooling water temperatures, were tested and analyzed. With the experimental data, a solar collector system was simulated on a larger scale, considering a 1 $m^2$ membrane. The membrane desorber evaluation shows that the desorption rate of water vapor increased as the LiBr solution temperature increased and the cooling water temperature decreased. Based on the experimental data from the membrane desorber/condenser, a theoretical heat load was calculated to size a solar system. Meteorological data from Emiliano Zapata in Mexico were considered. According to the numerical result, nine solar collectors with a total area of 37.4 $m^2$ provide a solar fraction of 0.797. The membrane desorber/condenser coupled to the solar system can provide an average of 16.8 kg/day of refrigerant fluid that can be used to produce a cooling effect in an absorption refrigerant system.

**Keywords:** membrane desorber; air gap membrane distillation; thermal solar energy; absorption cooling system

---

## 1. Introduction

Conventional vapor-compression air conditioning systems consume over 40% of the electricity and produce 24% of greenhouse gas emissions in the residential sector [1]. Research on sustainable energy options for air conditioning applications are an important issue. Among the prospective renewable energy solutions, the solar absorption cooling system is a suitable alternative because in summer, the cooling load increases due to increased solar radiation, enabling the coupling of solar thermal devices and cooling technologies driven by heat [2].

Several reports have been published about absorption cooling cycles driven by thermal solar energy. Lizarte et al. [3] analyzed a directly air-cooled absorber-condenser absorption chiller. This prototype used a $H_2O$/LiBr mixture and vacuum flat-plate collectors to provide thermal energy. The chiller capacity was 4.5 kW and a 42.2 $m^2$ solar collector system was necessary. The temperature of the storage tank was between 107 and 116 °C, with collector efficiency of 0.27. The desorber temperature was between 94.5 and 105 °C, and the evaporator temperature ranged from 25 to 15.5 °C. Ketjoy et al. [4] evaluated the performance of an absorption chiller that used $H_2O$/LiBr with a 35 kW cooling capacity. The system used heat pipe evacuated collectors with a total area of 72 $m^2$ and a gas backup system as the thermal source. The experimental results showed that the thermal efficiencies increased around 0.6 with solar irradiation values between 750 to 850 $W/m^2$. The average daily coefficient Of performance (COP) was 0.30, the maximum was 0.46 during summer, and the minimum was 0.11 registered during winter. Al-Ugla et al. [5] simulated three absorption air conditioning designs with $H_2O$/LiBr during 24-hour operation. Designs were based in different types of thermal energy storage. The simulation was programmed in Engineering Equation Solver (EES), and the operation temperatures were 84.6, 4.4, 38, and 36.2° C for the desorber, evaporator, condenser, and absorber, respectively. The thermodynamic analysis showed that the best alternative was integrating refrigerant storage with a solar absorption system due the collector size being calculated as 22.2 $m^2$. This was small compared to other designs; however, the maximum COP was 0.77. Bellos et al. [6] compared the results of the simulation of a solar absorption system with 250 kW cooling capacity, operating with $H_2O$/LiCl and $H_2O$/LiBr mixtures. The simulation was conducted in EES, and a flat plate collector field and three ambient temperatures (25, 30, and 35 °C) were considered. According to the results, the required area was between 587 and 689 $m^2$, the thermal efficiency was between 0.50 to 0.44, and COP was calculated as 0.83 to 0.85. Chen et al. [7] used TRNSYS to simulate an air conditioning system with 6 kW of cooling capacity using $H_2O$/LiBr. The simulation included 40 $m^2$ of evacuated glass tube solar collectors with a micro compound parabolic collector (CPC) reflector. The analysis showed an average thermal COP of 0.60 when the ambient temperature increased from 28 to 35 °C with a cooling capacity of 5.78 to 8.93 kW.

The integration of solar thermal energy into an absorption system is restricted by the desorber temperature; this component separates part of the refrigerant fluid from the refrigerant/absorbent liquid mixture [8]. The conventional boiling desorption process requires a constant heat load for the desorber, and for absorption systems that use water as the refrigerant fluid, vacuum pressure is required to separate the refrigerant fluid at low temperatures. An alternative to boiling desorption is membrane distillation (MD). In MD, the volatile component can be separated (in the vapor phase) from an aqueous working mixture by direct diffusion, so this process can occur below the boiling point. Thus, renewable thermal energy for powering the MD process for absorption cooling systems shows considerable potential.

Using membrane modules as desorbers for absorption heat pump applications has been widely studied and reported, and two configurations have been studied, namely hollow fiber membrane modules and flat membrane modules.

In the first category, there are the following reports: Hong et al. [9] proposed and analyzed a hydrophobic hollow fiber membrane (HFM) module for heat and mass transfer using numerical simulation to replace a conventional heat exchanger (called an economizer) in absorption cooling systems. The driving forces in this module are the differences in concentration and temperature between the hot concentrated solution and the cold diluted solution that flow inside the hollow fiber membranes and the shell side of the HFM device, respectively. Two configurations were simulated: direct contact and vacuum pressure. In the first case, the heat transfer via conduction through the HFM layers causes a decrease in the temperature difference between the two feed solutions, resulting in a decrease in the vapor mass transfer. In the second case, heat transfer via conduction can be considered negligible when the LiBr solution streams flow inside the hollow fibers and the shell side is filled with water vapor. Under this condition, the vapor mass transfer is higher. Hong et al. [10,11] provided an experimental and theoretical analysis of a HFM desorber. As solution temperature increased, the vapor pressure

increased exponentially, thereby enhancing the vapor mass transfer. The temperature polarization coefficient increased as the solution mass flux increased, indicating that the thermal difference between the solution channel and membrane decreased. The experimental desorption rate increased from 2.5 to 4 kg/m$^2$ h when the solution mass flux increased from 157.3 to 239.8 kg/m$^2$ s. The LiBr concentration in the solution was 51% *w/w*. The authors found that the vapor mass flux was enhanced when the condenser pressure was lower. The number of hollow fiber membranes can be adjusted for a specific volume of the module; therefore, a new lighter system is feasible.

In the literature, there are more reports about flat membrane modules: Venegas et al. [12] evaluated a membrane micro-desorber prototype operated with a heat source at 62 and 66 °C and H$_2$O/LiBr solution as the working mixture. The micro-desorber included a stainless steel plate with 50 rectangular microchannels 3 mm wide, 0.15 mm tall, and 58 mm long, and a polytetrafluoroethylene (PTFE) membrane with a 0.45 μm pore diameter. The desorption rate ranged from $1.60 \times 10^{-3}$ to $4.20 \times 10^{-3}$ kg/m$^2$ s. In another study, Venegas et al. [13] completed a parametric analysis of a membrane micro channel desorber working in direct diffusion and boiling modes with a H$_2$O/LiBr mixture. For membrane desorber design, the major parameters that must be considered include solution channel height and thickness of the heat exchange plate, regardless of operation mode. However, during operation, the inlet LiBr concentration restricts water vapor mass transfer in both cases. To maximize the desorption rate, the highest viable temperature of the thermal source must be selected. Venegas et al. [14] described a simplified model of a plate and frame membrane desorber with confined microchannels. It consisted of a hydrophobic membrane used to separate the water vapor and the H$_2$O/LiBr solution. The mean absolute error of the theoretical results was 26.4% compared with the experimental data. This result indicated the ability of the mathematical model to predict membrane desorber behavior.

Ibarra-Bahena et al. [15] proposed a new intermittent absorption cooling system including a membrane desorber/condenser coupled to a solar system. The membrane module was experimentally evaluated for four operation hours; after this time, the total amounts of refrigerant available, assuming a 1 m$^2$ membrane area, were 14.50, 11.59, and 7.20 kg with LiBr solution temperatures of 95.1, 85.2, and 75.1 °C, respectively. Based on these results, the intermittent solar absorption cooling system was calculated and sized. Ibarra-Bahena et al. [16,17] demonstrated the applicability of the air gap membrane distillation (AGMD) process for water vapor desorption. The authors used H$_2$O/LiBr and H$_2$O/Carrol mixtures, and all experimental tests were conducted at atmospheric pressure. The effects of aqueous solutions temperatures and concentrations were analyzed. The experimental desorption rate was 0.30–9.70 kg/m$^2$ h with the H$_2$O/LiBr mixture and 0.6–2.4 kg/m$^2$ h for the H$_2$O/Carrol mixture. Two membrane pore sizes (0.22 and 0.45 μm) were used, and the Carrol concentration was constant around 61% *w/w*.

Isfahani et al. [18] analyzed the effects of several operation variables on the desorption process in a membrane desorber. Based on the experimental results, the authors concluded that in direct desorption mode, the desorption rate behaves linearly with respect to the solution temperature. However, the desorption rate increases exponentially as the solution temperature increases in boiling desorption mode. The experimental results suggested that the solution pressure has no effect on the desorption process.

According to Bigham et al. [19], the water vapor desorption process can be enhanced if the solution layer at the membrane–solution interface is renewed. The authors proposed and evaluated a membrane desorber with microstructures in the flow channel to promote a turbulent flow regime; the desorption rate was increased around 1.7 times compared with the flow channel without the microstructures under the same operating conditions.

Wang et al. [20] proposed a vacuum membrane distillation process to replace the conventional boiling desorbers in absorption systems. The authors evaluated a 0.3 m$^2$ hollow fiber membrane desorber at different operation conditions. The maximum desorption rate was 2.0 kg/m$^2$ h with an 88 °C LiBr solution and 0.01 MPa vacuum pressure.

Thorud et al. [21] evaluated a membrane desorber with 0.00168 m$^2$ of effective area with a water/LiBr mixture. Based on the experimental data, the authors concluded that the desorption rate increased as the pressure difference increased, and the channel thickness and the LiBr concentration in the inlet stream decreased. Asfand and Bourouis [22] completed an extensive literature review of the use of membrane devices for absorption systems applications.

In this study, we evaluated a membrane desorber with an AGMD configuration with H$_2$O/LiBr as the working mixture, with solution temperatures from 75 to 95 °C, and condensation temperatures from 14 to 25 °C. With the experimental data, we calculated a thermal solar system and sizing to demonstrate the feasibility of a membrane desorber powered by renewable thermal energy for absorption cooling applications.

## 2. Air Gap Membrane Distillation Configuration

Membrane distillation is a process used to separate a volatile component present in a liquid mixture. The driving force is the temperature difference between both sides of the membrane, which causes a partial pressure difference. The most volatile component evaporates at the liquid–membrane interface, crosses through the porous membrane, and finally, condenses on the cold side. The key function of the membrane is to provide the liquid–vapor interface that is created on both sides. The hydrophobic membrane only allows the water vapor to cross [23]. In AGMD, a stagnant air gap exists between the membrane and the condensation surface. An AGMD module integrates a hot aqueous solution channel, an air gap section, and a cooling section. A cooling fluid flows on the other side of the cooling surface to remove the heat load delivered by the condensation process. The water vapor from the hot aqueous solution channel diffuses into the air gap, condenses on the cooling surface, and finally drains out. The AGMD process has two main advantages: lower heat losses and lower temperature polarization (TP). However, the air gap provides additional mass transfer resistance, resulting in a lower desorption rate. This mass transfer resistance is a function of the air gap width [24]. Figure 1 describes the AGMD process.

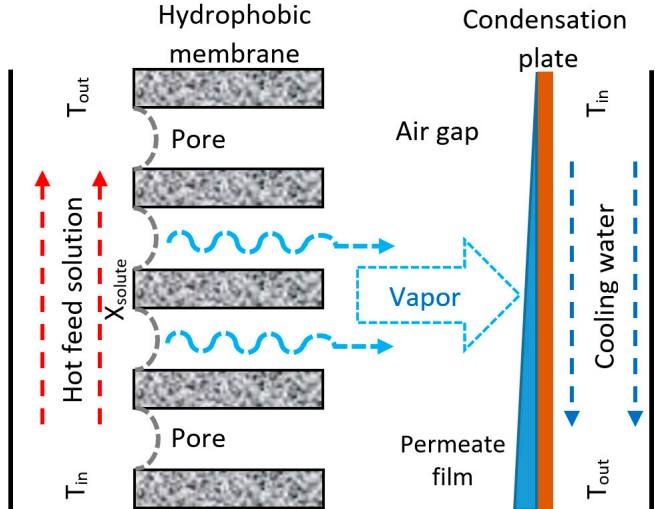

**Figure 1.** Air gap membrane distillation process schematic diagram.

## 3. Methodology

The membrane desorber/condenser module was evaluated under different thermal operating conditions. The H$_2$O/LiBr solution was selected because it is the most used working mixture and, generally, absorption cooling systems using the solution completed the desorption process at temperatures lower than 100 °C [25]. With the experimental data, a thermal solar energy system was calculated and sized to provide the heat load required for a desorber module with a 1 m$^2$ membrane

area. The details of the AGMD module and the simulation of the proposed solar system are described in the next subsections.

### 3.1. Membrane Desorber/Condenser Module

The experimental membrane module was constructed with two support plates, a hydrophobic membrane, a cooling plate, seal gaskets, and several bolts and screws. Support walls were composed of polymeric material with 25.4 mm thickness, and the gaskets were composed of neoprene and thermal-resistant silicon with 1 and 3 mm thickness, respectively. These gaskets were used to maintain a constant thickness of the flow channels and the air gap. A PTFE hydrophobic porous membrane with a 0.22 μm pore size and with a reported contact angle of 138° [26] was used. A metallic mesh was placed between the membrane and the air gap to reduce the membrane deformation caused by the LiBr solution flow. Figure 2 shows the components of the membrane desorber/condenser module. The silicon gasket formed a space between the support plate and the hydrophobic membrane, which was the $H_2O$/LiBr solution flow channel; it was 80 mm wide, 3 mm deep, and 180 mm long. An air gap with 3 mm of thickness was between the other side of the hydrophobic membrane, two neoprene gaskets, the metallic mesh, one silicon gasket, and one side of the condensing plate. Finally, the cooling water flows in the space between the other side of the condensing plate and the support plate. Previous studies described the experimental membrane desorber in detail [15–17].

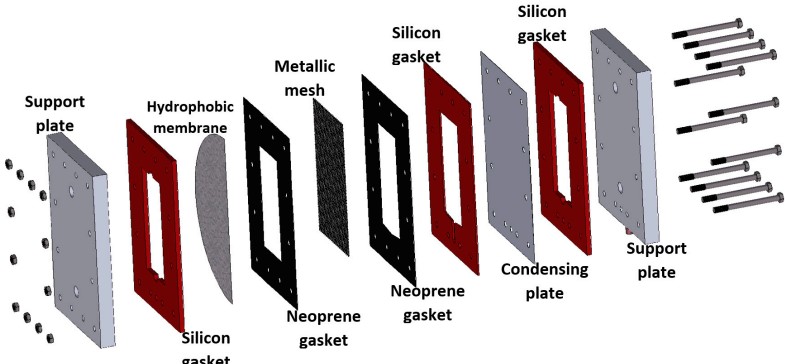

**Figure 2.** Membrane desorber/condenser module.

A 316 L SS plate heat exchanger (by Alfa Laval) was used to heat the $H_2O$/LiBr solution. A Cole-Parmer Polystat digital heating circulating bath with a 6 L capacity and 1000 W electrical power was used as the thermal source. A Halco CoolMax chiller (DCA504GUBB01 model) was used to chill water. Two storage tanks with 500 L capacities were used: one for storage of inlet cooling water for the AGMD module and the other for storing the cooling water after removal of the heat load by the condensation process inside the AGMD module. A Coriolis (by Micromotion) mass flow meter was used to measure the $H_2O$/LiBr solution mass flow rate. Two analogical flowmeters at the inlet of the plate heat exchanger and the membrane desorber were used to measure the volumetric flow rates of the heating fluid and the cooling water. A rubber septum was placed in the LiBr solution pipe, and a liquid sample of $H_2O$/LiBr was drawn out for each experimental test with a syringe. An ABBEMAT 200 (by Anton Paar) refractometer was used and, with the refractive index correlation reported previously for $H_2O$/LiBr solution, the LiBr concentration was calculated; the uncertainty of this correlation was ±0.15% *w/w* [27]. Two 32 W gear pumps were used to pump the solution and the heating fluid. The amount of distilled water was measured with an electronic balance. RTD pt100 temperature sensors were placed at the input and output ports of the membrane desorber and the plate heat exchanger (PHE). An Agilent data acquisition unit was used to record the temperature and the solution mass flow data. Figure 3 describes the experimental setup.

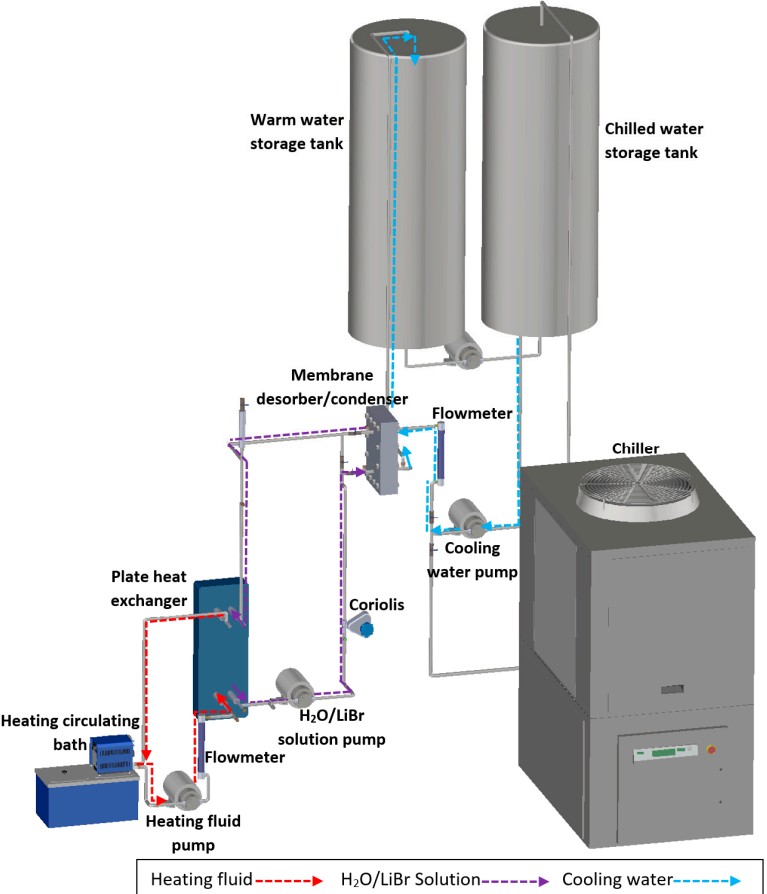

**Figure 3.** Schematic diagram of the experimental setup.

### 3.2. Experimental Conditions

Desorption rate was analyzed with different solution temperatures and cooling water temperatures under laboratory conditions and atmospheric pressure. The $H_2O$/LiBr solution mass flow rate ($m_{LiBr}$), and the cooling water ($V_{cw}$) and heating fluid ($V_{hf}$) volumetric flow rate for the initial LiBr concentration were kept constant at $2.50 \times 10^{-2} \pm 2.21 \times 10^{-5}$ kg/s, $1.2 \pm 0.05$ L/min, and $2.0 \pm 0.35$ L/min, respectively. In all experimental tests, the initial LiBr concentration ($X_{LiBr}$) was kept constant at $49.78\% \pm 0.03\%$ *w/w*. Table 1 provides the inlet and outlet temperatures of $H_2O$/LiBr solution ($T_{LiBr,in}$, $T_{LiBr,out}$, respectively) and cooling water ($T_{cw,in}$, $T_{cw,out}$, respectively) for each experimental test. The uncertainties of the measured variables are shown in Table 2.

**Table 1.** Experimental conditions.

| $T_{LiBr,in}$ (°C) | $T_{LiBr,out}$ (°C) | $T_{cw,in}$ (°C) | $T_{cw,out}$ (°C) |
|---|---|---|---|
| 75.3 | 73.6 | 25.1 | 25.8 |
| 75.2 | 73.5 | 21.9 | 22.6 |
| 75.3 | 73.5 | 19.5 | 20.3 |
| 75.3 | 73.7 | 16.6 | 17.6 |
| 75.3 | 73.3 | 14.4 | 15.6 |
| 80.2 | 78.3 | 25.4 | 26.0 |
| 80.2 | 78.3 | 21.6 | 22.5 |
| 80.2 | 78.2 | 19.6 | 20.5 |
| 80.5 | 78.5 | 17.0 | 18.2 |
| 80.2 | 78.0 | 14.5 | 15.7 |
| 85.2 | 83.0 | 25.3 | 26.1 |
| 85.2 | 82.8 | 22.1 | 23.1 |

**Table 1.** *Cont.*

| $T_{LiBr,in}$ (°C) | $T_{LiBr,out}$ (°C) | $T_{cw,in}$ (°C) | $T_{cw,out}$ (°C) |
|---|---|---|---|
| 85.3 | 83.1 | 20.0 | 21.0 |
| 85.2 | 83.0 | 16.8 | 18.0 |
| 85.3 | 82.7 | 14.5 | 15.9 |
| 90.3 | 87.8 | 25.1 | 26.1 |
| 90.2 | 87.6 | 21.9 | 23.1 |
| 90.2 | 87.8 | 19.6 | 20.9 |
| 90.1 | 87.6 | 16.9 | 18.3 |
| 90.3 | 87.5 | 14.7 | 16.3 |
| 95.2 | 92.5 | 24.9 | 26.0 |
| 95.2 | 92.3 | 21.8 | 23.0 |
| 95.2 | 92.2 | 19.9 | 21.2 |
| 95.2 | 92.2 | 17.1 | 18.5 |
| 95.2 | 92.1 | 14.5 | 16.2 |

**Table 2.** Uncertainty of the measured variables.

| Variable | Sensor/Instrument | Operation Range | Uncertainty |
|---|---|---|---|
| Temperature ($T$) | RTD PT100 | −30 to 350 °C | ± 0.2 °C |
| Volumetric flow ($V_{cw}$) | Volumetric flowmeter | 0 to 7 L/min | ± 5.0% f.s. * |
| Volumetric flow ($V_{hf}$) | Volumetric flowmeter | 0 to 1.2 L/min | ± 4.0% f.s. * |
| Mass flow ($m_{LiBr}$) | Coriolis mass flowmeter | 0 to $4.0 \times 10^{-2}$ kg/s | ± 0.1% |
| Distillate water weight ($w_{dis}$) | Electronic balance | 0 to 600 g | ± 0.01 g |
| Refractive index ($IR$) | Electronic Refractometer | 1.3000 to 1.7200 | ± 0.0001 |

Note: * f.s., full scale.

### 3.3. Mathematical Model

A mathematical model, described in detail previously [16], was used to calculate the thermal energy requirement and desorption rate by a membrane desorber/condenser, considering a 1 m² membrane area. The thermodynamic properties of the H$_2$O/LiBr solution were estimated for each experimental condition with mathematical correlations reported previously [28–30].

### 3.4. Solar System Simulation

The solar system was sized based on a theoretical desorption heat load. An Apricus model AP-30 solar collector was experimentally tested, and its thermal efficiency ($\eta_i$) curve was calculated using ASHRAE 93-1986 standard [31] as follows:

$$\eta_i = 0.485 - 1.579 \left( \frac{\Delta T}{G_b} \right) \tag{1}$$

where $\Delta T$ is the outlet and inlet temperature difference (°C), and $G_b$ is the solar radiation (W/m²).

Based on this thermal efficiency, the solar system was simulated and sized for the desorber energy requirements. An insulated storage tank was considered, and serial configuration was assumed for the solar system simulation. The energy balance inside the storage tank is defined as follows:

$$\left( mC_p \right)_t \frac{dT}{dt} = Q_u - Q_{loss} - Q_{Des} \tag{2}$$

where $Q_u$ is the useful heat load, $Q_{loss}$ is the storage tank heat loss, and $Q_{Des}$ is the heat load required by the membrane desorber. The temperature inside the storage tank ($T^+_t$) was calculated using Equation (3) solved by finite differences, as follows:

$$T^+_t = T_t + \frac{\Delta t}{\left( mC_p \right)} [Q_u - Q_{loss} - Q_{Des}]. \tag{3}$$

## 4. Results

### 4.1. Experimental Desorption Rate

Figure 4 shows that as the solution temperature increased, the desorption rate increased. This behavior was non-linear because the vapor pressure exponentially increased as the solution temperature increased, meaning an exponential increase in the driving force. As the cooling water temperature decreased, the desorption rate increased. This effect occurred because a decrease in the cooling water temperature led to an increase in the vapor pressure difference across the membrane; thus, the desorption rate was enhanced [32]. The highest desorption rate was 5.69 kg/m$^2$ h at 95.2 °C LiBr solution and 14.6 °C cooling water; the lowest desorption rate value was 1.53 kg/m$^2$ h with a 75.6 °C LiBr solution and 25.1 °C cooling water. As is shown in Figure 4, the water desorption from aqueous LiBr solution using the AGMD process could occur within wide thermal operating conditions at atmospheric pressure. This means that renewable thermal energy sources, like solar energy, could be used to power the desorption process. In addition, according with this figure, the effect of the H$_2$O/LiBr solution temperature was higher than the condensation temperature, i.e., when the solution temperature was 75.3 °C (average), the desorption rate increased from 1.53 kg/m$^2$ h to 2.26 kg/m$^2$ h, with the cooling water temperature of 25.1 °C and 14.4 °C, respectively, which means an increment of 0.48 times. On the other hand, when the cooling water temperature was 25.2 °C (average), the desorption rate increased from 1.53 kg/m$^2$ h to 5.03 kg/m$^2$ h, with the solution temperature of 75.3 °C and 95.2 °C, respectively, which represents an increase of 2.3 times. This behavior has been reported in the literature; according with Khalifa et al. [33], the effect of the coolant temperature on the vapor desorption is not as significant as the hot saline solution temperature and air gap width. For conventional boiling desorbers, the influence of the condenser temperature is similar [34,35].

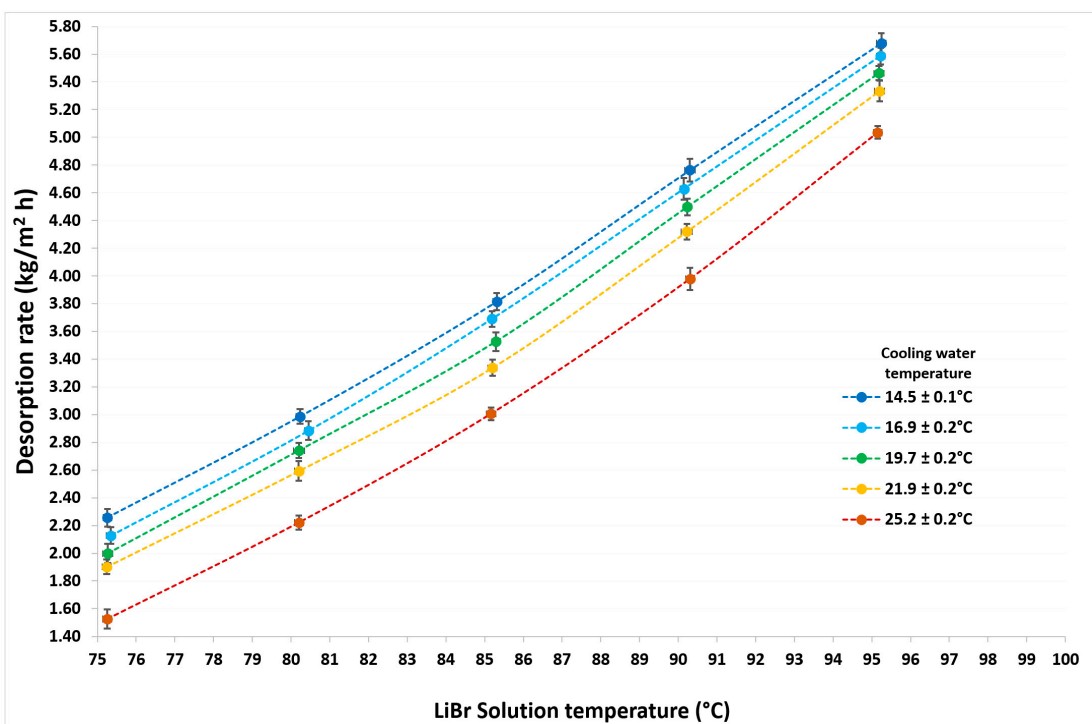

**Figure 4.** Desorption rate as a function of LiBr solution and cooling water temperatures.

### 4.2. Theoretical Desorption Rate

The experimental and theoretical desorption rates are compared in Figure 5. The maximum deviation of the mathematical model with respect to the experimental data was 6.89%. This mathematical model has been reported and validated in a previous paper [16], and it was based on the AGMD separation process [36,37]; however, the $H_2O$/LiBr solution transport properties, such as viscosity, limited the mass transfer process. Therefore, the membrane desorption rate can be improved by renewing the solution–membrane boundary layer [19].

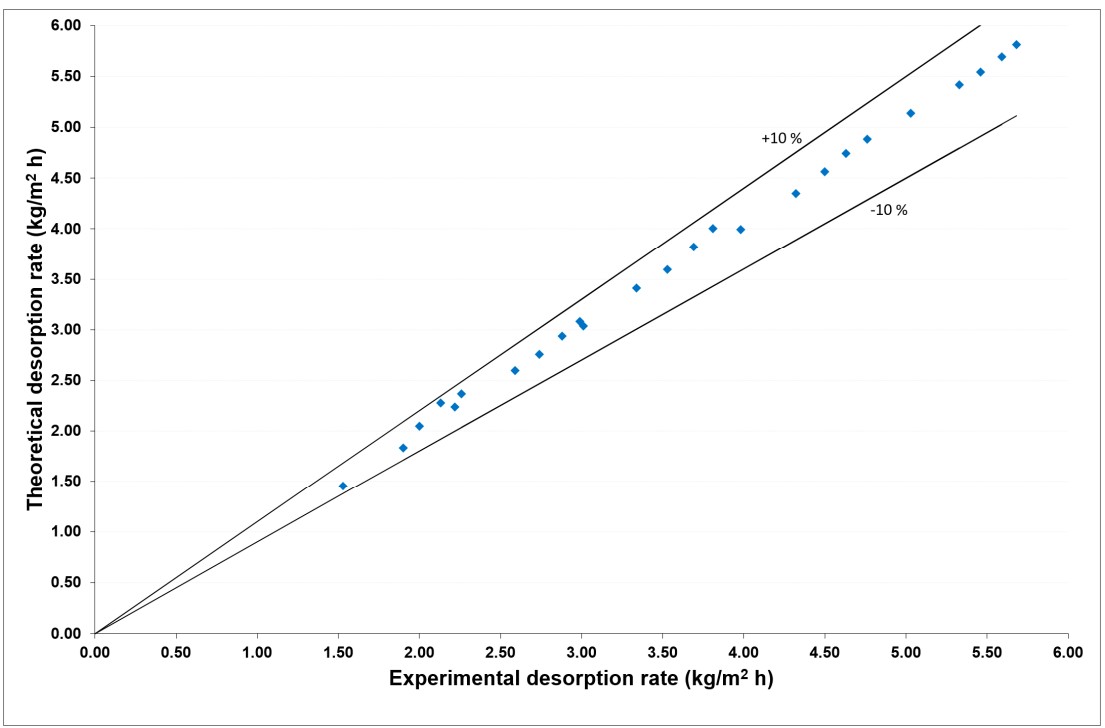

**Figure 5.** Theoretical versus experimental desorption rate.

The validation of the mathematical model with the experimental data allowed us to calculate the thermal heat load required for a large-scale membrane desorber/condenser. As mentioned in Section 2, the driving force of the AGMD process is the temperature difference across the membrane. Figure 6 depicts the desorption rate as a function of the average temperature difference between the $H_2O$/LiBr solution and the cooling water ($T_{LiBr,av} - T_{cw,av}$). These temperatures were defined as follows:

$$T_{LiBr,av} = \frac{T_{LiBr,in} - T_{LiBr,out}}{2} \qquad (4)$$

$$T_{cw,av} = \frac{T_{cw,in} - T_{cw,out}}{2} \qquad (5)$$

In this figure, the discontinued line represents the theoretical trend and is compared with the experimental data; it can be seen that the desorption rate increased with the temperature difference. For the mathematical model, we assumed all refrigerant vapor (water vapor) diffused through the stagnant air gap was totally condensed (distillate water) on the condense plate, meaning that the distillate water in the vapor phase could not leave the device, even by the only small drain at the bottom because it was sealed continuously by a water drop. For the output temperatures, the deviation was less than 5% for the cooling water and 1.8% for the $H_2O$/LiBr solution on average.

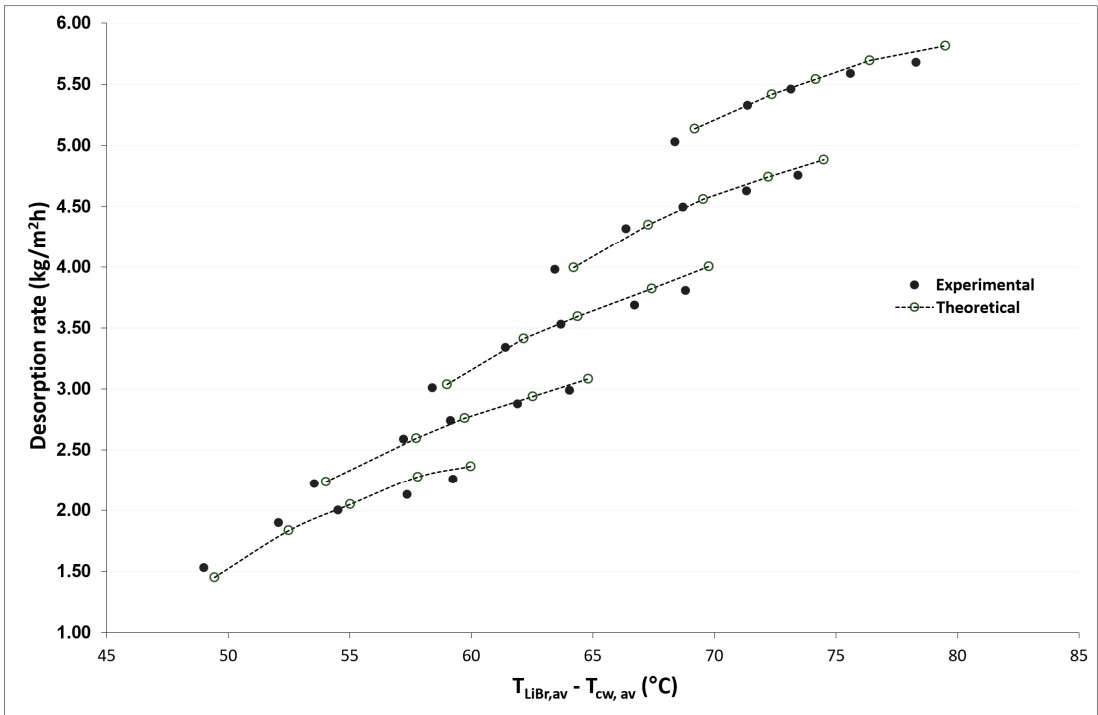

**Figure 6.** Desorption rate as a function of the average temperature difference between the hot feed solution and cooling water.

### 4.3. Solar System

For simulation and sizing of the solar thermal system, we used a numerical algorithm [38]. We considered the meteorological data for a typical year in Emiliano Zapata, Morelos, Mexico. In this city, the average ambient temperature is around 27.6 °C from 09:00 to 17:00; this temperature value is similar to the experimental cooling water temperature tested, so this value was considered in the AGMD thermal load calculation. Figure 7 shows the solar radiation throughout one year. The solar thermal system proposed was designed for a maximum heat load of 10 kW for eight hours of operation. This means that the solar thermal system had to supply 2000 L of hot water per day to keep the temperature range of the $H_2O/LiBr$ solution within 75 to 95 °C. A 2000 L storage hot water tank was considered that delivers 80 kWh of thermal load using a heat exchanger with a 0.85 thermal effectiveness [39]. Figure 8 depicts the thermal water profile inside the storage tank (*Ts*) and the ambient temperature (*Ta*). On some days, the water temperature could be higher than 100 °C. This was possible because the solar system was pressurized; therefore, no phase change would occur. According to the numerical results, nine Apricus model AP-30 evacuated tube solar collectors with a total area of 37.4 m$^2$ are necessary to provide a solar fraction of 0.7973. However, the remaining energy can be supplied by a conventional auxiliary system to ensure a constant supply of hot water at the working temperature required by the desorption process.

The theoretical result showed that the membrane desorber/condenser module can produce an average of 16.8 kg/per day of refrigerant fluid (distilled water). However, for cloudy days when the solar radiation is less than 600 W/m$^2$, the water temperature inside the storage tank does not reach the minimum operating temperature of 75 °C; this condition occurs 49 days (average) per year. According with Figure 8, on 49 days (average) per year, the storage tank may not reach the minimum operation temperature.

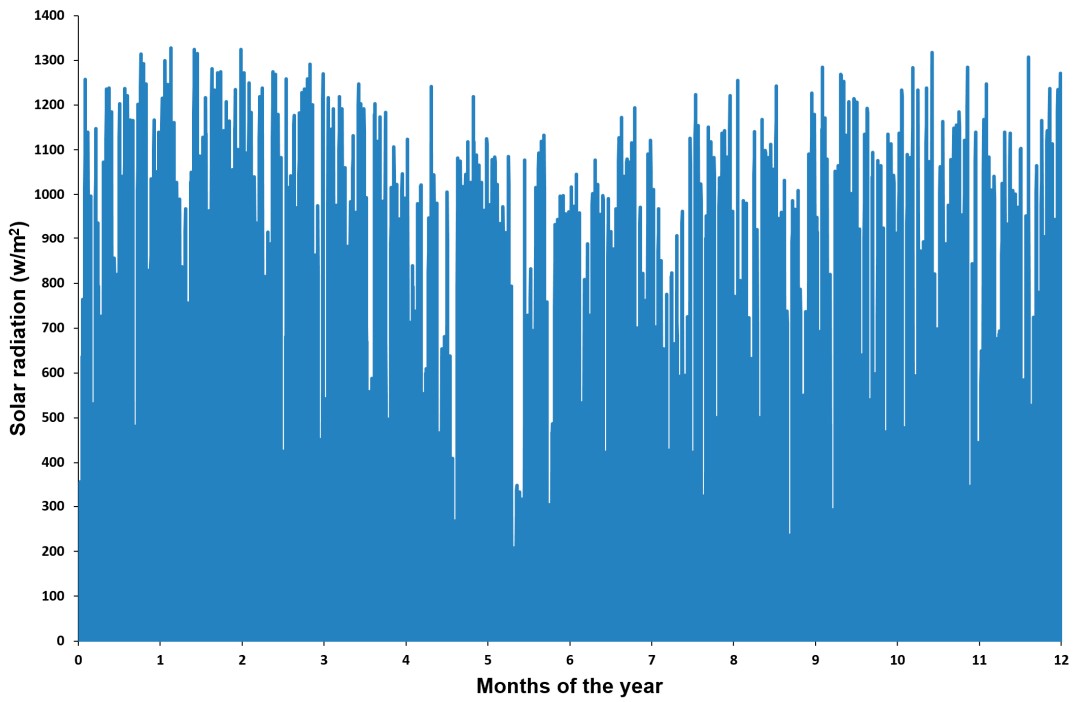

**Figure 7.** Solar radiation along a year in Emiliano Zapata city, Morelos, México.

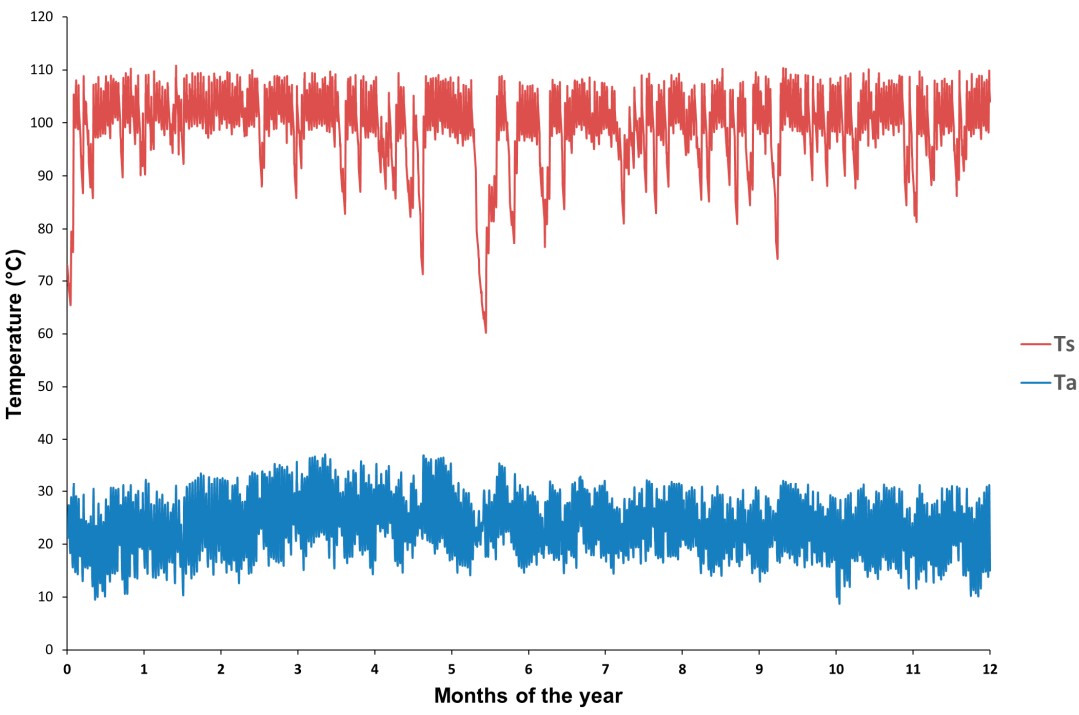

**Figure 8.** Water thermal profile and ambient temperature.

## 5. Conclusions

In this study, we evaluated a membrane desorber/condenser with an AGMD configuration and a $H_2O$/LiBr solution. The LiBr solution and cooling water temperatures were tested and analyzed. Based on the experimental results, we found the desorption rate of water vapor increases with increasing LiBr solution temperature and decreasing cooling water temperature. This phenomenon occurs because as the temperature difference between both sides of the membrane increases, so does the vapor pressure



difference across the membrane. However, LiBr solution transport properties, such as viscosity, limit the vapor mass transfer across the solution layer compared with the conventional membrane desalination process. The highest desorption rate was 5.69 kg/m$^2$/h with a 95.2 °C LiBr solution and 14.6 °C cooling water; the lowest desorption rate value was 1.53 kg/m$^2$/h with a 75.6 °C LiBr solution and 25.1 °C cooling water. With the experimental data, a solar collector system was simulated on a larger scale with a 1 m$^2$ membrane area using the meteorological data from Emiliano Zapata in Mexico. Based on the numerical results, nine solar collectors with a total area of 37.4 m$^2$ provide a solar fraction of 0.797. The membrane desorber/condenser coupled to the solar system can provide an average of 16.8 kg/day of refrigerant fluid that can be used to produce a cooling effect in an absorption refrigerant system. This result demonstrates the feasibility of the membrane desorber/condenser driven by thermal solar energy for absorption cooling system applications.

**Author Contributions:** Conceptualization, J.I.-B., E.V.-R., Y.R.G.-L., W.R., R.J.R., and U.D.-C.; funding acquisition, E.V.-R., W.R., R.J.R., and A.R.-M.; investigation, J.I.-B., E.V.-R., Y.R.G.-L., U.D.-C., and W.R.; methodology, J.I.-B., E.V.-R., Y.R.G.-L., W.R., R.J.R., A.R.-M., and U.D.-C.; validation, J.I.-B., E.V.-R., Y.R.G.-L., U.D.-C., and W.R.; writing—original draft, J.I.-B., E.V.-R., Y.R.G.-L., W.R., and U.D.-C.; writing—review and editing, J.I.-B., E.V.-R., Y.R.G.-L., W.R., R.J.R., A.R.-M., and U.D.-C. All authors have read and agreed to the published version of the manuscript.

**Funding:** This work was funded by the projects PRODEP, FORDECYT 297486, and PAPIIT-UNAM IT100920.

**Acknowledgments:** We thank the PRODEP, FORDECYT 297486, and PAPIIT-UNAM IT100920 projects. Yuridiana R. Galindo-Luna thanks DGAPA-UNAM for the postdoctoral fellowship. Jonathan Ibarra-Bahena thanks the CIC-2019 UNAM postdoctoral fellowship. Ulises Dehesa-Carrasco thanks Cátedras-CONACYT-México project 1772.

**Conflicts of Interest:** The authors declare no conflict of interest.

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
