# Peer review of "Feasibility Analysis of a Membrane Desorber Powered by Thermal Solar Energy for Absorption Cooling Systems"

_applsci, doi:10.3390/app10031110_

Round 1

Reviewer 1 Report

Line 38-69: 1 paragraph can be and should be divided into few smaller paragraphs for the ease of reading.

Other paragraphs could also be divided into shorter parts

Line 118: no comma

Table 1: The concentration in all the cases is in fact the same.

Table 2: What is the uncertainty of LiBr/H2O concentration measurement/calculation (not the refractive index)?

Line 229L How was the solar radiation measured? What was the position of solar collector for the efficiency measurement?

Line 250-251: “The maximum deviation of the mathematical model from the mathematical model was 6.89.” What does it mean?

Figure 5: What are those different lines?

Line 299-300: Could you provide the average number of those days compared to the number of days when the temperature is higher than 75?

Author Response

Answer to reviewers

Reviewer 1

Thanks for your comments and support

Line 38-69: 1 paragraph can be and should be divided into few smaller paragraphs for the ease of reading.

It was corrected

Other paragraphs could also be divided into shorter parts

It was corrected

Line 118: no comma

It was corrected

Table 1: The concentration in all the cases is in fact the same.

Yes it is, we reported the concentration values for each experimental test in order to demonstrate that the concentration was kept constant.

Table 2: What is the uncertainty of LiBr/H2O concentration measurement/calculation (not the refractive index)?

The error of the correlation was added

Line 229L How was the solar radiation measured? What was the position of solar collector for the efficiency measurement?

Solar collector was tested using ASHRAE 93-1986 standard. According with this standard, the radiation must be measured on the apparent plane of the collection surface. Therefore, a solar radiation measure device was placed in the same plane of collector surface. In order to keep the perpendicular plane of the angle of incidence of solar radiation, the collector was placed on a mobile test surface.

Line 250-251: “The maximum deviation of the mathematical model from the mathematical model was 6.89.” What does it mean?

It was corrected

The sentence refers to the accuracy of the theoretical model respect to the experimental results which must be expressed in percent (%).

Figure 5: What are those different lines?

Represents the trend of calculated theoretical data

Line 299-300: Could you provide the average number of those days compared to the number of days when the temperature is higher than 75?

A comment about this was added (lines 341 and 342)

Reviewer 2 Report

The manuscript "Feasibility Analysis of a Membrane Desorber Powered by Thermal Solar Energy for Absorption Cooling Systems"  analysis  the membrane desorber using AGMD configuration with LiBr aqueous solution. The theoritical calculation was supported by using experimental data.

The manuscript is well defined and interesting.

I would suggest to publish after some minor changes:

The supplier of the PTFE membrane; What is the contact angle of used PTFE membrane (since it was mentioned as hydrophobic)? The experimental membrane desorber should be explained more instead of showing only reference [15-17] Discussion should be extended and supported by literature.

Author Response

Reviewer 2

R: Thanks for your comments and support

The manuscript "Feasibility Analysis of a Membrane Desorber Powered by Thermal Solar Energy for Absorption Cooling Systems” analysis the membrane desorber using AGMD configuration with LiBr aqueous solution. The theoritical calculation was supported by using experimental data.

The manuscript is well defined and interesting.

I would suggest to publish after some minor changes:

The supplier of the PTFE membrane; What is the contact angle of used PTFE membrane (since it was mentioned as hydrophobic)? The experimental membrane desorber should be explained more instead of showing only reference [15-17] Discussion should be extended and supported by literature.

R: The supplier of the PTFE membrane did not provide us the contact angle, and we did not measure this parameter, however, in literature, the contact angle for this membrane type is around 138°

R: The experimental membrane desorber/condenser module description was extended and a depict figure (Figure 2) was added.

R: A discussion about the cooling water effect was included in section 4.1.

Reviewer 3 Report

This paper presents a combination of experimental and simulation results about a solar cooling facility integrated by evacuated tube thermal collectors and an absorption chiller. The generator of the chiller uses membrane technology, specifically it is an air gap membrane distillation module. The paper is well written and results are very interesting for the scientific and technical community. The following comments have been derived from the review:

- Fig. 2: Please, increase the size of the names and include arrows to indicate the flows direction.

- Eq. (1): Subscript of G is not the same used in line 229.

- Eq. (3): DeltaT refers to the time increment, for this reason a lowercase “t” should be used.

- Line 251: Percentage symbol is missing.

- Does the temperature differences represented in Fig. 5 correspond to the average of inlet values?

- Figs. 6 and 7: Please use the same units in the horizontal axis of both figures.

- Temperature of 27.6 ºC is something higher than values used in the experimental study (14.4 - 25.4 ºC). Please, comment about the influence of this parameter on the desorber performance and the area of collectors required.

- Why different values appear for the storage tank volume, 2000 and 2100 L? How this volume was calculated?

Author Response

Reviewer 3

R: Thanks for your comments and support

This paper presents a combination of experimental and simulation results about a solar cooling facility integrated by evacuated tube thermal collectors and an absorption chiller. The generator of the chiller uses membrane technology, specifically it is an air gap membrane distillation module. The paper is well written and results are very interesting for the scientific and technical community. The following comments have been derived from the review:

- Fig. 2: Please, increase the size of the names and include arrows to indicate the flows direction.

R: Figure 2 was improve.

- Eq. (1): Subscript of G is not the same used in line 229.

R: It was corrected.

- Eq. (3): DeltaT refers to the time increment, for this reason a lowercase “t” should be used.

R: It was corrected.

- Line 251: Percentage symbol is missing.

R: It was corrected.

- Does the temperature differences represented in Fig. 5 correspond to the average of inlet values?

R: The temperature differences between inlet and outlet streams (of H2O/LiBr solution and cooling water) are in X-axis in Fig. 5. Equations 4 and 5 were added in order to clarify this.

- Figs. 6 and 7: Please use the same units in the horizontal axis of both figures.

R: It was corrected.

- Temperature of 27.6 °C is something higher than values used in the experimental study (14.4 - 25.4 °C). Please, comment about the influence of this parameter on the desorber performance and the area of collectors required.

R: A discussion about the cooling water effect was included in section 4.1. The area of collectors were calculated assumed 27.6°C

- Why different values appear for the storage tank volume, 2000 and 2100 L? How this volume was calculated?

R: It was corrected.

R: In section 4.3 we explain the storage tank calculation. “The solar thermal system proposed was designed for a maximum heat load of 10 kW for eight hours of operation. This means that the solar thermal system must supply 2000 L of hot water per day to keep the temperature range of the H2O/LiBr solution within 75 to 95 ° C.”

Round 2

Reviewer 1 Report

Why is the concentration in table 1 shown? It does not affact anything, as it is basically the same in all rows.

line 287-295 yes, it is normal fact in the abosrption chillers.

line 373-374 needs language revision: in this form it isn't easily understandable.

Author Response

Thanks for your comments

Why is the concentration in table 1 shown? It does not affact anything, as it is basically the same in all rows.

LiBr concentration column was delete in table 1, an average LiBr concentration value was included.

line 287-295 yes, it is normal fact in the abosrption chillers.

Two references were included in order to support this sentence.

line 373-374 needs language revision: in this form it isn't easily understandable.

These lines were improve.

Reviewer 2 Report

The review done very well.

I would suggest to publish this paper.

Author Response

Thanks for your support for publication

Reviewer 3 Report

All the previous comments have been adequately taken into account by the authors. The paper is recommended for publication in the journal.

Author Response

Thanks for your support for publication